# Do Summarization Models Synthesize?

## Abstract

Multi-document summarization entails producing concise synopses of collections of inputs. For some applications, the synopsis should accurately *synthesize* inputs with respect to a key property or aspect. For example, a synopsis of film reviews all written about a particular movie should reflect the average critic consensus. As a more consequential example, consider narrative summaries that accompany biomedical *systematic reviews* of clinical trial results. These narratives should fairly summarize the potentially conflicting results from individual trials.

In this paper we ask: To what extent do modern multi-document summarization models implicitly perform this type of synthesis? To assess this we perform a suite of experiments that probe the degree to which conditional generation models trained for summarization using standard methods yield outputs that appropriately synthesize inputs. We find that existing models do partially perform synthesis, but do so imperfectly. In particular, they are over-sensitive to changes in input ordering and under-sensitive to changes in input compositions (e.g., the ratio of positive to negative movie reviews). We propose a simple, general method for improving model synthesis capabilities by generating an explicitly diverse set of candidate outputs, and then selecting from these the string best aligned with the expected aggregate measure for the inputs, or *abstaining* when the model produces no good candidate. This approach improves model synthesis performance. Our hope is that by highlighting the need for synthesis (in some summarization settings), this work motivates further research into multi-document summarization methods and learning objectives that explicitly account for the need to synthesize.

## 1 Introduction

*Multi-document summarization* (MDS) models aim to distill inputs into concise synopses that preserve key content. Examples of MDS include summarizing news articles (Dang, 2005; Fabbri et al., 2019; Ghalandari et al., 2020; Evans et al., 2004), answering questions from multiple sources (Dang, 2006), and producing overviews of scientific literature (Liu et al., 2018; Lu et al., 2020; Mollá & Santiago-Martínez, 2012; Wallace et al., 2020; DeYoung et al., 2021). We expect summarization models to produce outputs consistent with inputs (Kryscinski et al., 2020; Nan et al., 2021a), e.g., discussing the same types of entities (Nan et al., 2021b) and allowing one to answer questions similar in a way that is consistent with individual inputs (Wang et al., 2020a; Scialom et al., 2021).

In some applications models must *synthesize* inputs—i.e., aggregate potentially conflicting information—to yield an accurate synopsis (Figure 1). As a simple example, consider the meta-reviews of movies featured on Rotten Tomatoes,[1] which provide a consensus view of individual critic opinions. These reviews should therefore reflect the mean and range of sentiment implicit in the input critiques: A summary of mostly negative reviews (e.g., *Gigli*) should communicate that the film was widely panned; a summary of mixed reviews (as in the case of *The Fifth Element*) ought to convey that critics disagreed and discuss the main positive and negative attributes.

A more consequential example is the task of summarizing the evidence presented in clinical trials. Individual trials will frequently present conflicting evidence about whether or not a particular health intervention is effective. An ideal summary of the evidence would appropriately weigh the findings presented in the constituent inputs and reflect the evidence on balance.

---

[1]A website that aggregates film reviews: `https://www.rottentomatoes.com/`.

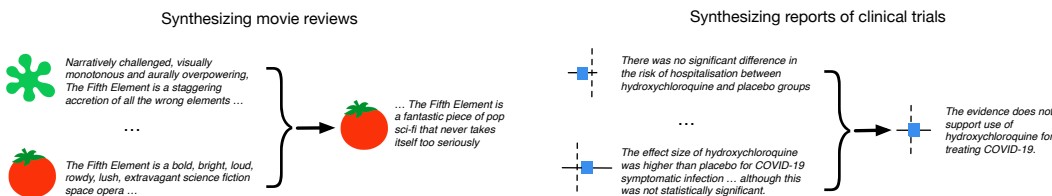

Figure 1: Two multi-document summarization tasks where models must implicitly synthesize inputs to produce accurate summaries. Left: Summarizing film reviews with varying sentiment to yield a *critics consensus*. Right: Summarizing trials that have evaluated a particular medical invention.

What are the desiderata of multi-document *synthesis*? First, summaries produced by models should be *consistent* with the input data, with respect to the latent property of interest. In the case of Rotten Tomatoes, the *sentiment* of the summary should be in line with the aggregate sentiment expressed in the individual critic reviews. A corollary to this is that models should be *sensitive* to changes in the composition of inputs, e.g., removing most of the negative reviews from a set of inputs should yield a summary with a corresponding increase in the expressed sentiment.

In this work we evaluate neural MDS models with respect to these criteria. To this end we use a meta-reviews dataset from Rotten Tomatoes (Leone, 2020) and a dataset of systematic reviews (meta-analyses) summarizing the evidence about medical interventions (Wallace et al., 2020). For the former we probe the degree to which generated meta-review sentiment agrees with the expected aggregate sentiment score; for the latter we evaluate whether the generated summary indicates that the input evidence suggests, on balance, that the intervention under consideration was effective.

Our **main contributions** are summarized as follows. (1) To the best of our knowledge, this is the first work to investigate implicit *synthesis* in summarization, and the degree to which modern models are capable of this.[2] (2) We show that "off-the-shelf" neural MDS models are somewhat inconsistent and insensitive with respect to performing synthesis in summarization. (3) We propose and evaluate a simple and general technique which involves generating a diverse set of output candidates (Vijayakumar et al., 2016) and then selecting from these on the basis of agreement with an expected aggregate measure (based on inputs), with promising results.

## 2 SYNTHESIS AND SUMMARIZATION

In standard multi-document summarization, we assume inputs $(X_i, y_i)$, where $X_i = \{x_{i1}, ..., x_{i|X_i|}\}$. We then typically train a summarization model with parameters $\theta$, to consume $X_i$ and yield summaries $\hat{y}_i$ as similar as possible to targets $y_i$. More precisely, the standard objective entails finding estimates for $\theta$ which maximize target token log-probabilities. Assuming the input documents $x_{ij}$ in $X_i$ have been linearized (i.e., concatenated, usually with adjoining special tokens to demarcate individual inputs) into a string $x_i^{\oplus}$ of input tokens, this objective takes the form: $\sum_{t=1}^{|y_i|} \log p_\theta(y_{it}|y_{i1}, ..., y_{i(t-1)}, x_i^{\oplus})$, where $p_\theta$ is a probability assigned to the token at position $t$ in the (linearized) target $x_i^{\oplus}$ by a summarization model with parameters $\theta$. By myopically focusing on encouraging the model to produce tokens that mimic the targets, this objective aligns with standard (but flawed) measures of automated summary quality like ROUGE (Lin, 2004), which quantify $n$-gram overlap between targets $y_i$ and outputs $\hat{y}_i$.

We are interested in settings in which there is an additional, latent property implicit in the constituent input texts $x_{ij}$, $z_{ij}$. For example, $z_{ij}$ might reflect the sentiment in critique $j$ of the film indexed by $i$. Summaries should *synthesize* this aspect, i.e., the generated summary $\hat{y}_i$ should implicitly convey an aggregated $z_i$ which reflects a synthesis or aggregation $G$ over $Z_i = \{z_{i1}, ...z_{i|X_i|}\}$. That is, we assume $z_i = G(Z_i)$ . In both cases considered here—summaries of film critiques and synopses of clinical trials evidence—$G$ can reasonably be assumed to be a (weighted) mean, $G(Z_i) = \frac{1}{|X_i|} \sum_{j=1}^{|X_i|} \alpha_{ij} z_{ij}$. That is, summaries should roughly reflect the average sentiment and reported treatment effect in the cases of movie reviews and clinical trial reports, respectively.

---

[2]See Appendix B for related content aggregation work, over structured relations Shah et al. (2021a).

|  | Train | Dev | Test | Train | Dev[†] | Test |
|---|---|---|---|---|---|---|
| Number of metareviews | 7251 | 932 | 912 | 1675 | 360 | 397 |
| Avg. metareview length | 32.0 | 32.6 | 32.4 | 101 | 107 | 111 |
| Total number of inputs | 195033 | 24336 | 24474 | 11054 | 1238 | 2669 |
| Avg. number of inputs | 26.9 | 26.1 | 26.8 | 6.6 | 3.4 | 6.7 |
| Avg length of individual input | 30.6 | 30.8 | 30.6 | 475 | 379 | 449 |
| Avg length of concatenated inputs | 822 | 804 | 822 | 2641 | 1336 | 2544 |
| Target Percent Positive | 59.5 | 62.1 | 61.2 | 31.9 | 31.4 | 35.0 |

Table 1: Dataset statistics for movie reviews (left) and systematic reviews (right). Number of meta-reviews, average meta-review length (tokens), number of input reviews per split, average number of inputs per instance, average total length of an input to an instance. For movie reviews, the target percent positive reports the fraction of metareviews with a positive sentiment; for systematic reviews this refers to the fraction of metareviews reporting a significant effect. † We subset the original dev set to instances of $\leq 4k$ tokens (to accommodate T5; the other models can consume up to 16k).

We investigate the following questions. (1) Do model summaries $\hat{y}_i$ reflect the anticipated aggregate aspect of interest? That is, how well calibrated is the aspect communicated in the generated summary ($z_{i\hat{y}}$) compared to the expected $z_i$? (2) Can we *improve* the ability of summarization models to synthesize by explicitly incorporating synthesis targets $z_i$ into the decoding process?

We propose a simple inference-time procedure to explicitly preference output candidates that align with the expected aggregate property of interest (e.g., average sentiment), and report promising results for the approach. This strategy also naturally lends itself to *cautious* summarization, i.e., approaches in which we allow the model to *abstain* from generating an output if it does not produce any candidates that reflect the anticipated aggregate measure.

## 3 DATASETS AND MEASUREMENTS

### 3.1 MOVIE REVIEWS

We first consider a dataset comprising movie reviews and associated meta-reviews summarizing these from Rotten Tomatoes. An in-house staffer summarizes audience reviews [3] into meta-reviews. These meta-reviews synthesize the constituent input reviews, and reflect the aggregate critic reception of a film. Each meta-review is associated with a numerical "Tomatometer" score, which is an overall measure of what percent reviews were positive for the corresponding film ($G$ then is an average of the positive indicator per review). The Rotten Tomatoes dataset we use comprises 9095 movies with meta-reviews constructed from 244,000 individual reviews (Table 1).

**Measuring sentiment in movie reviews.** As our measure $g$ we train a BERT model (Devlin et al., 2019) using the continuous (fine-grained) sentiment targets provided in the SST dataset (Socher et al., 2013).[4] We trained this model for 3 epochs using a learning rate of 5e-5 using the `Huggingface` library[5] with no hyperparameter tuning. While the raw text of the SST dataset is in-domain, the targets themselves are not. We find a reasonably strong correlation between our sentiment estimates and the "true" meta-review sentiment ("Tomatometer" score): The $R^2$ (centered) is 0.696, mean squared error (MSE) of 0.022, and Pearson's r of 0.836 (Figure 2, upper left).

### 3.2 BIOMEDICAL SYSTEMATIC REVIEWS OF TREATMENTS

Our second dataset is a collection of systematic reviews from the Cochrane Collaboration.[6] This dataset comprises roughly 2600 systematic reviews summarizing a total of 16,500 clinical trials evaluating interventions in healthcare (Table 1). Each review includes both a natural language sum-

---

[3] written by designated "top-critics", audience members recognized for quality and quantity of reviews

[4] SST is itself based on a collection of Rotten Tomatoes critic reviews (Pang & Lee, 2005). We verified that the SST text fragments do not overlap with any of our target reviews by manually checking any (fragment, review) pair with substantial ($>= 75\%$) overlap for approximately one quarter of all reviews.

[5] https://github.com/huggingface/transformers/blob/main/examples/pytorch/text-classification/run_glue.py

[6] An international non-profit dedicated to helping healthcare providers make evidence-based decisions.

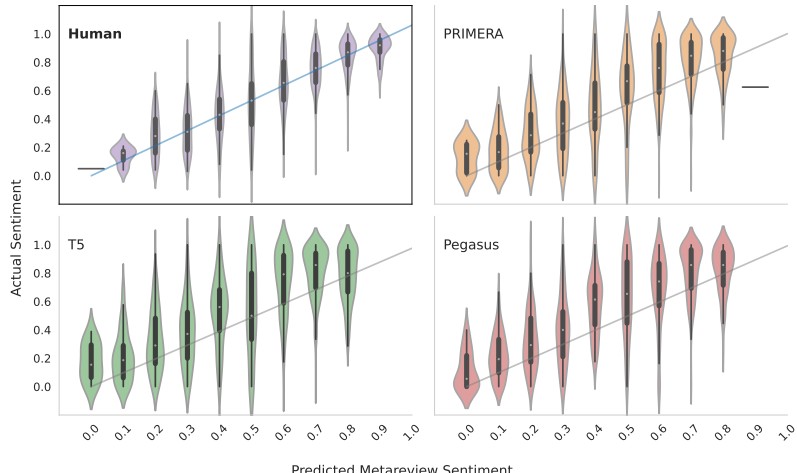

Figure 2: Movie Reviews: Actual vs. Predicted Sentiments on generated summaries. We replaced LED with human outputs (upper left) for comparison; see Figure 8 in Appendix D for all models.

mary and accompanying statistical meta-analysis results. The latter provides an aggregate statistical summary of the individual (study-level) data extracted from the trials included in each review. The natural language summary should accurately convey and contextualize the findings of the meta-analysis. Therefore, the (lack of) treatment efficacy communicated in a given summary should generally agree with the direction of the corresponding meta-analytic point estimate.

**Measuring effects in evidence syntheses** For systematic reviews of clinical trials, we resort to a less granular *classification* model $g(x_{ij}), g(y_i)$ which attempts to infer whether a given piece of text reports a significant result or not. In particular we use `RobotReviewer` (Marshall et al., 2017; DeYoung et al., 2020). Given a narrative describing a clinical trial result (or a systematic review summary of such results), `RobotReviewer` predicts whether the reported result indicates a significant effect of the treatment being investigated, or not. We can compare this prediction to the "truth", which here is derived from the meta-analytic result (specifically by checking whether $p < 0.05$). Applying this off-the-shelf model to the manually composed summaries accompanying the meta-analyses in our Cochrane set, we observe a macro-average F1 score of 0.577 (Table 10, Appendix D), providing a reasonable (if weak) measure for this task.

## 4 MODELS

We evaluate a suite of transformer (Vaswani et al., 2017) summarization models: Longformer (Beltagy et al., 2020), Pegasus (Zhang et al., 2020), PRIMERA (Xiao et al., 2021), and T5 (Raffel et al., 2020). PRIMERA was designed and pre-trained specifically for multi-document summarization specifically. And while not explicitly designed as multi-document summarization models, both Pegasus Zhang et al. (2020) and T5[7] have been used on multi-document tasks, while Longformer has been used for a related multi-document summarization task (DeYoung et al., 2021). For all models we mostly use hyperparameters defaulted to in their respective `huggingface` implementations. We conduct a hyperparameter sweep over optimization steps and learning rate, selecting the best model by ROUGE1 performance on the dev set (Appendix C, Tables 8. 9).

## 5 EXPERIMENTS

### 5.1 HOW WELL DO SUMMARIZATION MODELS SYNTHESIZE?

We report sentiment performance for all models in Table 2. These are metrics quantifying the strength of the relationship between (a) the continuous sentiment inferred (via our text-regression

---

[7]https://huggingface.co/osama7/t5-summarization-multinews

| | $R^2$ | Pearson's r | MSE | ROUGE1 |
|---|---|---|---|---|
| LED | 0.551 | 0.742 | 0.042 | 0.242 |
| PRIMERA | 0.608 | 0.780 | 0.037 | 0.254 |
| T5 | 0.516 | 0.720 | 0.046 | 0.253 |
| Pegasus | 0.530 | 0.730 | 0.044 | 0.245 |
| Reference | **0.697** | **0.836** | **0.023** | |

| | F1-score | ROUGE1 |
|---|---|---|
| LED | 0.490 | 0.259 |
| PRIMERA | 0.526 | 0.253 |
| T5 | 0.521 | 0.206 |
| Pegasus | 0.568 | 0.212 |
| Reference | **0.577** | |

Table 2: Base synthesis results. **Movie reviews** (left): correlations between sentiment measured in model outputs and target sentiments. We report $R^2$, Pearson's r, and mean-squared errors. **Systematic reviews** (right): we report macro-averaged F1s. ROUGE1 included for reference.

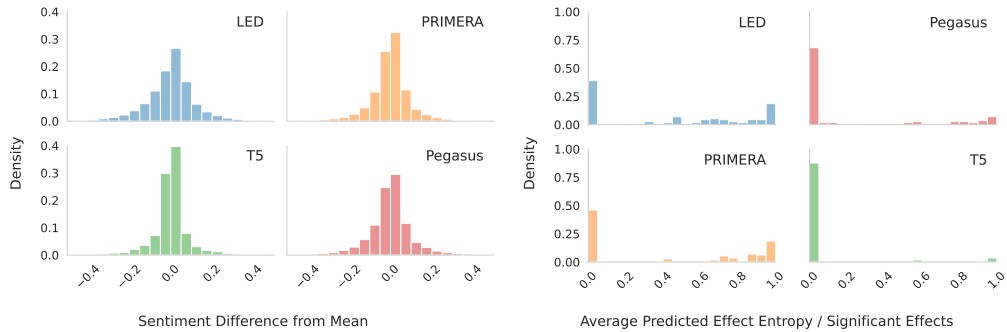

Figure 3: The spread of sentiment/treatment effect measured in outputs produced from permuted input orderings. Left: Movie review sentiment. Right: Systematic review significance prediction entropy (0 indicates order insensitivity) on the subset of reviews that report *significant* effects.

$g$) over model generated or reference (human written) summaries and (b) the reference sentiment (Tomatometer) score. Across these metrics, correlations between the sentiment measured in model generated outputs and the Tomatometer score are considerably lower than that between the same measurement over human-composed summaries and said score. Based on these metrics, human authors do a better job of synthesis than the models when composing their summaries.

For systematic reviews (Section 3.2), we are able to measure $g$ whether a text appears to report significant treatment effect or not, and we can compare this against the $p$-value from the corresponding statistical meta-analysis. This permits only a coarse assessment of synthesis, as we are unable to measure correlations. Instead we report classification metrics describing how often the effect significance inferred from a summary (generated or manually written) matches the ground truth derived from the meta-analysis (Table 2). The results are qualitatively similar to the sentiment case, in that the humans appear to do a better job of synthesis — as best we can measure, the significance reported in their summaries better aligns with the statistical results than in model generated summaries.

## 5.2 SENSITIVITY TO INPUT ORDERING

Synthesis of inputs should be invariant to ordering (e.g., the critics' consensus on a film does not depend on the order in which one reads the reviews). Here we evaluate if models are sensitive to input orderings with respect to the synthesized aspect of interest ($z_{i\hat{y}}$) in the resultant outputs. Specifically, $X_i = \{x_{i1}, ..., x_{i|X_i|}\}$ will constitute an arbitrary ordering of inputs reflected in the linearized version $x_i^{\oplus}$. This ordering should not affect the aggregate aspect $z_{i\hat{y}}$ in the summary.

To evaluate if models realize this invariance, we permute the instance $i$ inputs $X_i$ (and, consequently, the linearized $x_i^{\oplus}$) one hundred times, randomizing input orderings. For each such permutation $\tilde{X}_i$ (and associated $\tilde{x}_i^{\oplus}$), we generate a summary $\hat{y}_i$ and estimate of the resultant aspect $\tilde{z}_{i\hat{y}}$, using the corresponding measurement model. By repeating this process for each instance $i$, we can construct an empirical distribution over $\tilde{z}_{i\hat{y}}$'s under different random orderings.

**Movie reviews.** We zero-mean the $\tilde{z}_{i\hat{y}}$'s inferred over each instance, and combine the distributions from all instances into a histogram (Figure 3 left). This shows the spread of sentiments inferred over outputs under random input orderings minus the corresponding instance mean sentiment. Were

|         | $R^2$ | Pearson's r | MSE   |
|---------|-------|-------------|-------|
| LED     | 0.524 | 0.724       | 0.057 |
| PRIMERA | 0.572 | 0.756       | 0.052 |
| T5      | 0.481 | 0.694       | 0.063 |
| Pegasus | 0.499 | 0.706       | 0.060 |

|         | F1-score | Accuracy |
|---------|----------|----------|
| LED     | 0.510    | 0.684    |
| PRIMERA | 0.533    | 0.680    |
| T5      | 0.469    | 0.675    |
| Pegasus | 0.452    | 0.658    |

Table 3: **Movie reviews** (left): Correlation between subsampled inputs and generated meta-reviews. **Systematic reviews** (right): macro-averaged results (F1 and accuracy) for subsampled inputs.

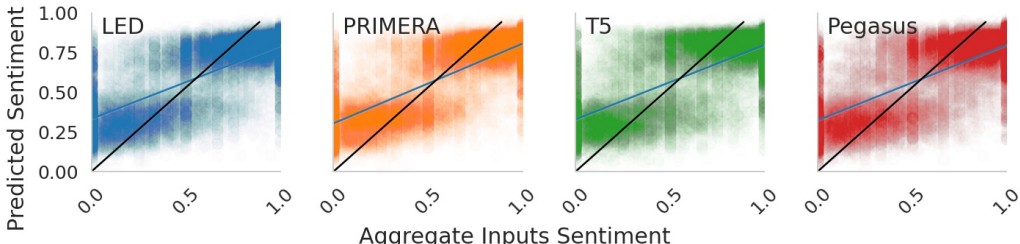

Figure 4: Model sentiment sensitivity to manipulated input sentiment. The intensity patterns indicate that models tend to oscillate between low and high sentiments in outputs, and are not responsive to subtler shifts in input sentiment compositions. For context we include a model regression (blue) and the reference sensitivity regression (black).

a model completely invariant to ordering, the empirical distribution over these differences would collapse to 0. Instead, we observe a relatively wide spread in the sentiment measured over outputs generated from different permutations, indicating a counter-intuitive sensitivity to orderings.[8]

**Systematic reviews**. For each $X_i$ we have 100 order permutations and associated summaries; we infer whether these report *significant results* or not, and record the fraction that do ($p_i$). If models were invariant to ordering, this fraction would always be 0 or 1. Values in-between suggest the model flips the report conclusion as a result of different input orderings. We calculate the entropy of $p_i$ to quantify this. Figure 3 (right) shows a histogram of these entropies calculated over the subset of examples where the associated meta-analysis indicates a significant effect.[9] Densities away from zero indicate sensitivity to ordering.

## 5.3 SENSITIVITY TO INPUT COMPOSITION

Synthesis models should be responsive to changes in the distribution of the attribute to be synthesized in the input composition: If we increase the ratio of positive to negative reviews in an input set, we would anticipate a concomitant change in the sentiment communicated in the meta-review $z_{i\hat{y}}$. To assess if models meet this synthesis desiderata, we manipulate model inputs $X_i$ in such a way as to induce an expected change in the target measure $z_{i\hat{y}}$; we then measure if the output yields a summary that aligns with this expected change.

**Movie reviews**. We manipulate the ratio of positive to negative reviews and observe the resultant change in the property of interest latent in the corresponding output. We take movies with mixed reviews, and delete 10%, 20%, 30%, ..., 100% of the positive inputs, retaining the negative inputs; we then repeat the process but instead remove negative inputs. For each of these permutations, we measure the input sentiment, the meta-review sentiment, and how well they correlate (Table 3).

Figure 4 plots the relationship between the fraction of positive reviews in the (manipulated) input sets and the granular sentiment score inferred over the resultant outputs. The models are generally undersensitive to changes in their input: rather than having a change in meta-review sentiment equivalent in size to changes in input sentiment (a slope of 1, as we observe when we fit a model to the human written summaries). Models tend to have trouble changing their sentiment, and require a large change in input distribution to substantially change the sentiment communicated in the output.

---

[8]For a ROUGE1 comparison, see Appendix E, Figure 10.

[9]These are the more interesting cases; we provide results over the entire dataset in Appendix Figure 9.

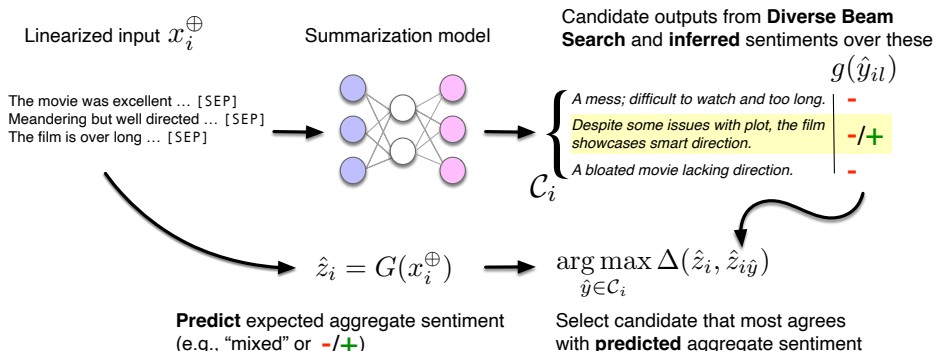

Figure 5: Proposed strategy to improve synthesis. We generate an intentionally diverse set of output candidates (Vijayakumar et al., 2016) and then select from these the text that best agrees with the *predicted* aggregate property of interest (here, sentiment). We can also *abstain* when the model fails to yield an appropriate output.

**Systematic Reviews**. To measure sensitivity to changes in input composition, we manipulate our inputs $X_i$ such that the meta-analysis result (target $z_{i\hat{y}}$) flips from a significant effect to no effect, or from no effect to an effect. Operationally, we do this by first taking of a subset of the reviews that have conflicting evidence (yielding 139 unique reviews). We then order inputs in these by (weighted) effect sizes,[10] and remove subsets which ought to flip the significance result.

## 6    IMPROVING SYNTHESIS IN SUMMARIZATION

We propose a simple post-hoc approach to improving the synthesis performed by multi-document summarization models. This involves the following steps: (1) Generate an explicitly *diverse* set of output candidates[11]; (2) Select from these as the final output the candidate that best agrees with the expected synthesis result (as predicted by an external model).[12]

For (1), we rely on a previously proposed technique for generating diverse outputs $\mathcal{C}_i$ from input $x_i^\oplus$, namely *Diverse Beam Search* (DBS) (Vijayakumar et al., 2016). This method modifies standard beam search to maintain multiple *groups* of beams. During decoding, a term is added to the next-token log probabilities which effectively penalizes production of (partial) strings similar to candidates on beams in *other* groups.[13]

In (2) we would like to select the output that best synthesizes the property of interest; this requires a mechanism for specifying what we *expect* the synthesized property be, given the inputs. For example, if we know the sentiment scores associated with input movie reviews, we might enforce that the sentiment expressed in the output agrees with the average of these. To realize this intuition, we can select as final output from $\mathcal{C}_i$ the string that best aligns with this anticipated aggregate property (sentiment score or significance finding). Operationally, this requires an external model to measure—or estimate—the aspect of interest as latent in a given candidate output. This is a limitation of the approach, but in many settings it may be feasible to identify or construct a model; we were able to do so for both tasks considered in this paper.

There is no guarantee that *any* member of $\mathcal{C}_i$ will align well with the anticipated aggregated property. In such cases, we have no means of yielding an output consistent with respect to synthesis, and it may be desirable to *abstain* from outputting anything at all in such cases; that is, to be a *cautious* summarizer (Ferri et al., 2004; Hechtlinger et al., 2018). We consider this strategy in the case of generating narrative synopses of evidence, as this constitutes a case in which (a) one would very

---

[10]In fixed effects meta-analysis the weights are inverse variances associated with study-level effect estimates.

[11]See Appendix Tables 11, 12 for an ablation over diversity vs. standard beam search outputs

[12]For a related generate-and-select approach Oved & Levy (2021) see Appendix B.

[13]This penalty is associated with a hyperparameter $\lambda$ that encodes the relative importance of realizing diverse; we use $\lambda$=0.5 here and did not extensively tune this. Other hyperparameters include number of groups and total number of beams; we used 5 for both of these, retaining 5 beams as used for analysis above.

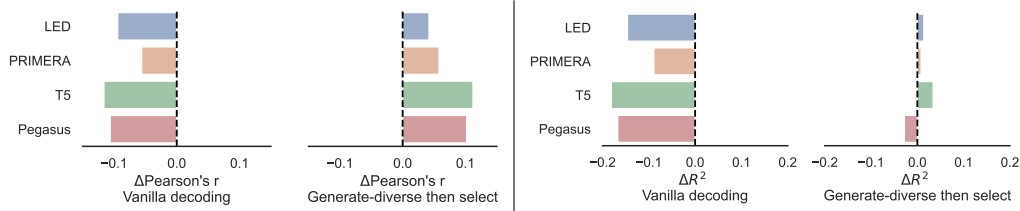

Figure 6: Differences relative to human summaries under vanilla decoding and the proposed generate-diverse then select strategy on the Rotten Tomatoes dataset and task. We report Pearson's r and $R^2$, both measures of synthesis "calibration". Vanilla decoding yields synthesis performance worse than humans, but explicitly considering synthesis at inference time as proposed results in performance comparable to and sometimes better than the human summaries (as best we can measure).

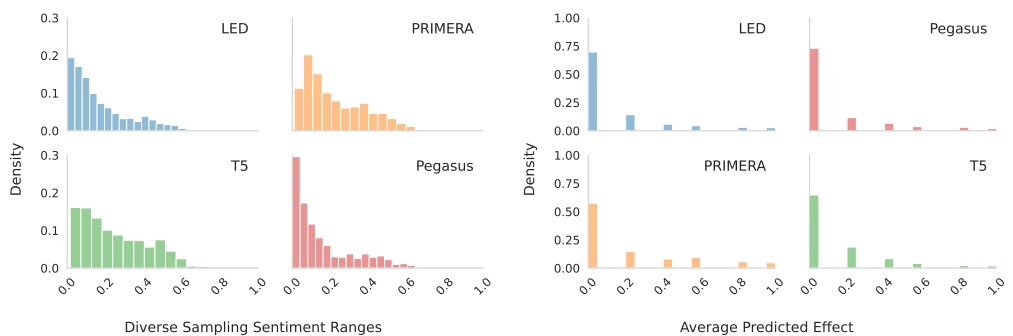

Figure 7: Distributions of outputs for the candiate summaries. **Movie reviews** (left) show a histogram for the range of differences between lowest and highest output sentiments. **Systematic reviews** (right) show histograms of the fractions of outputs reporting *significant* results.

much prefer not to produce a misleading summary of clinical evidence (Kell et al., 2021), and, (b) we observe many cases where the diverse decoding strategy yields an output that seems to communicate (at a granular level) the aggregate findings expected.

**Movie Reviews** For movie reviews we use a BERT (Devlin et al., 2019) model trained on IMDB (Maas et al., 2011)[14] to predict the sentiment of each input $x_{ij}$, using the proportion of $x_{ij} \in X_i$ with a positive score as an approximation for the target sentiment $z_{i\hat{y}}$. For each diverse prediction $\mathcal{C}_i$, we predict a sentiment $\tilde{z}_{i\hat{y}}$ using our sentiment regression model (Section 3.1), and select the prediction cloest to the estimated target sentiment $|\tilde{z}_{i\hat{y}} - z_{i\hat{y}}|$. We find this improves model performance to human-like levels in terms of synthesis, as best we can measure (Table 4, Figure 6).

**Systematic Reviews**. In the case of systematic reviews, we can have only a binary measure of *significant effect* (or not). As a proxy for $z_{i\hat{y}}$, we again use RobotReviewer to extract an effect for each of the model inputs $x_{ij}$, using the majority vote (i.e., do the plurality of $x_{ij} \in X_i$ indicate that there was an effect). We classify each output candidate in $\mathcal{C}_i$ again using RobotReviewer to estimate $\tilde{z}_{i\hat{y}}$. We then select for output the highest probability candidate in $\mathcal{C}_i$ which agrees with the majority vote of the inputs, and abstain where there are no viable candidates. For the models we do choose a summary for, we find performance similar to our measure (Table 5). Movie reviews show a wide range of sentiments; systematic reviews show some improvement but are biased towards no effect (qualitatively observed in Appendix G).

# 7    RELATED WORK

**Automatic (multi-document) summarization** (Nenkova & McKeown, 2011; Maybury, 1999) has been an active subfield within NLP for decades. We have focused our analysis on modern, neural abstractive models for conditional text generation (Bahdanau et al., 2015). In light of their empirical

---

[14]https://huggingface.co/lvwerra/bert-imdb

| | $R^2$ | MSE | Pearson's r | R1 | | $R^2$ | MSE | Pearson's r | R1 |
|---|---|---|---|---|---|---|---|---|---|
| LED | 0.656 | 0.032 | 0.821 | 0.229 | LED | 0.763 | 0.022 | 0.878 | 0.227 |
| Pegasus | 0.694 | 0.029 | 0.835 | 0.229 | Pegasus | 0.799 | 0.019 | 0.894 | 0.232 |
| PRIMERA | 0.749 | 0.024 | 0.880 | 0.240 | PRIMERA | 0.890 | 0.011 | 0.948 | 0.240 |
| T5 | 0.721 | 0.026 | 0.856 | 0.231 | T5 | 0.876 | 0.012 | 0.938 | 0.230 |
| Reference | 0.697 | 0.023 | 0.836 | | Reference | 0.697 | 0.023 | 0.836 | |

Table 4: Movie Reviews: Generate diverse movie meta-reviews and then choose among them using an approximate target sentiment (left) or the oracle sentiment (right). R1 is ROUGE1 score.

| | F1 | %Abstention | ROUGE1 | Abstention-Oracle | ROUGE1-Oracle |
|---|---|---|---|---|---|
| LED | 0.557 | 0.386 | 0.252 | 0.233 | 0.259 |
| PRIMERA | 0.581 | 0.336 | 0.251 | 0.213 | 0.248 |
| T5 | 0.568 | 0.350 | 0.202 | 0.228 | 0.210 |
| Pegasus | 0.588 | 0.383 | 0.211 | 0.242 | 0.225 |
| Reference | 0.577 | | | | |

Table 5: Systematic Review results with modified-then-selected predictions. F1 is a macro-averaged F1 on the set of returned results. We abstain when no output matches the expected synthesis result.

success, we have specifically evaluated a set of Transformer-based (Vaswani et al., 2017) models which have recently been used for multi-document summarization (Beltagy et al., 2020; Zhang et al., 2020; Xiao et al., 2021; Raffel et al., 2020). There has been some work on highlighting conflicting evidence in health literature specifically (Shah et al., 2021b;a), though this was focused primarily on highlighting conflicting evidence, and explicitly aggregating extracted content.

**Sentence fusion** One view on synthesis might be that is a particular kind of *sentence fusion* (Barzilay & McKeown, 2005). However, past work on "fusing" sentences has assumed that the aim is to generate an output that contains the information common to similar sentences (Thadani & McKeown, 2013). This is intuitive in the context of, e.g., summarizing multiple news articles covering the same event. But here we are interested in the more challenging setting in which the output should reflect an aggregate measure of potentially conflicting evidence or opinions.

**Interpretation and analysis of neural models for NLP** This work is also related to the emerging body of work on analyzing neural NLP models, their behaviors, "knowledge", and "abilities" in general e.g., Linzen et al. (2016); Tenney et al. (2019); Petroni et al. (2019); Niven & Kao (2019); Meng et al. (2022). There has been some work specifically on analyzing neural summarization models. Xu et al. (2020a) investigated when a model is likely to extract (copy) rather than abstract (generate). Xu & Durrett (2021) furthered this analysis by assessing when models were relying on the local input to produce particular output tokens, and when they instead rely on mostly on a background language distribution acquired in pre-training.

**Factuality of neural summarizers** Neural conditional generation models have proven adept at producing fluent outputs, but in the context of summarization they are prone to *hallucinating* content unsupported by input documents (Maynez et al., 2020; Kryscinski et al., 2019). Automated metrics such as ROUGE do not reliably capture such phenomena (Falke et al., 2019; Maynez et al., 2020). This has motivated several efforts to design automated factuality metrics (e.g., Wang et al. (2020b); Xu et al. (2020b); see Pagnoni et al. (2021) for an overview).

## 8 CONCLUSIONS

We have outlined and investigated the problem of *synthesis* as related to some summarization tasks. We showed that existing models are partially able to synthesize implicitly, but do so imperfectly: For instance, the aggregation they perform is sensitive to input ordering, and they are not as sensitive to perturbations in the composition of inputs as one would hope. We proposed and validated a straightforward inference time method to improve model synthesis capabilities by preferentially outputting summary candidates that align with a predicted aggregate measure, and demonstrated empirically that this offers gains in performance. Our hope is that this work encourages additional research into summarization models that explicitly optimize to accurately synthesize potentially conflicting evidence and information.

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

## A    NOTATION

| Variable | Definition |
|---|---|
| $X_i$ | The $i^{th}$ set of input documents, corresponding to instance $i$ |
| $y_i$ | The target summary $y$ of the $i^{th}$ instance |
| $y_{i,t}$ | The $t^{th}$ token of target $y_i$ |
| $\hat{y}_i$ | A generated summary of the $i^{th}$ instance |
| $x_{ij}$ | The $j^{th}$ input document for instance $i$ |
| $x_i^{\oplus}$ | A particular linearization of the input documents $X_i$ |
| $\theta$ | Model & parameters |
| $p_\theta$ | Probably under parameters $\theta$ |
| $p_\theta(y_{i,t}|y_{i,1..t-1}, x_i^{\oplus})$ | Standard auto-regressive prediction of the next token given an input and partial summary |
| $z_{ij}$ | Latent property (sentiment, significance finding) of $x_{ij}$ |
| $z_i$ | Aggregated latent property (sentiment, significance finding) of $X_i$ |
| $z_{i\hat{y}}$ | Latent property measured over summary $\hat{y}$ |
| $G$ | Aggregation function over latent properties $z_{ij}$, yields $z_i$ |
| $g$ | Auxillary function to measure latent property $z_{ij}$ of $x_{ij}$, or $z_{i\hat{y}}$ of $\hat{y}$ |
| $\alpha_{ij}$ | A weight for $z_{ij}$ |

Table 6: Notation.

## B    ADDITIONAL RELATED WORK

**Shah et al. (2021a)** created a "nutri-bullets" system for generating consensus-based summaries of health and nutrition related content. They assume a low-supervision setting in which one has a set of tuples extracted from texts with which to train a content extractor, and where one can design heuristic rule-based aggregation strategies on top of extracted tuples mapping onto discrete categories like "consensus". By contrast, we assume a more typical supervised summarization setting and are interested in continuous aggregation of a latent attribute of interest, and we do not assume (or have access to) relational tuples over inputs. Indeed, recent work Wolhandler et al. (2022) has shown that systematic reviews are categorically different than news summarization, and that relational tuple extractors do not perform well in the medical domain.

First, Shah et al. (2021a)'s focus primarily on settings in which training data is (severely) limited, and motivate their pipeline approach on the basis of this limited supervision assumption. For this reason they define separate modules: The first performs content selection (tuple extraction; this does require manual annotations of tuples on a subset of texts to train such an extractor); The second applies (manually composed) deterministic aggregation rules over these extracted tuples to combine them; a final module then generates a "surface realization" conditioned on the aggregated result.

We have investigated more typical supervised settings (with thousands of input and summary pairs), and we are training modern end-to-end transformer-based summarization models. We have empirically assessed the extent to which model outputs in this typical training regime are consistent with the continuous synthesis result anticipated. We do not have annotated tuples on our inputs (as would be required to use the Shah et al. (2021a) approach, as it assumes a trained content extractor module). And while applying discrete (manually composed) aggregation operators over inputs makes sense in some settings, we are explicitly interested in the ability of models to aggregate variables of interest continuously, for example producing "very positive" summaries when movie reviews are overwhelmingly positive, and merely "positive" summaries when they are only mostly positive.

In sum, the approach proposed by Shah et al. (2021a) is appropriate in, and designed for, low-supervision settings (which we do not consider here) where there are natural "tuples" to be extracted from inputs and supervision for this sub-task (which we do not have) and where discrete aggregation categories of inputs is natural (whereas we are interested in continuous aggregation, e.g., mean sentiment).

**Wolhandler et al. (2022)** attempts to measure how challenging multi-document summarization is, as a function of the unique knowledge (represented as relational tuples) required to produce a summary.

| Model | Huggingface Checkpoint | Optimizer | Schedule | Warmup | Smoothing |
|-------|------------------------|-----------|----------|--------|-----------|
| LED | allenai/led-base-16384 | Adam | P.nomial/decay 0.01 | 50 steps | 0.1 |
| T5 | t5-base | Adam | Linear | 50 steps | 0 |
| PRIMERA | allenai/PRIMERA | Adam | Linear | 50 steps | 0 |
| Pegasus | sshleifer/distill-pegasus-cnn-16-4 | Adafactor | Linear | 500 steps | 0.1 |

Table 7: Model hyperparameters. We used optimizers, schedulers, weight decay, and label smoothing as best according to examples from source implementations (where available). Optimizer warmup was arbitrarily chosen. Non-specified parameters were the Huggingface defaults.

This work measures how many new tuples each input document might add in *contrast* to subsets of other inputs. By greedily building subsets of inputs as a function of new information added, they find that standard multiple document summarization datasets merely need to select two to four documents from inputs of up to ten, whereas their approach breaks down in the case of systematic reviews. They find that due to both technical constraints for relation extraction, as well as the inability to model contradiction, relational extraction and aggregation methods are insufficient for producing evidence syntheses.

**Oved & Levy (2021)** introduce the Perturb and Selection Summarizer (PASS) system for summarizing Amazon product reviews. It works by perturbing model inputs (i.e. keep random subsets of the input), generating a summary for each perturbation (via standard beam search), and then selecting amongst outputs (via a ranker) to produce a coherent, self-consistent, and fluent summary.

PASS is similar to our work in that it generates multiple outputs and selects amongst them. However it differs in several key respects. The key conceptual difference between PASS and our work is that PASS's target is a summary's self-consistency (a product review might contradict itself on some aspect, e.g. simultaneously discussing a product fitting well in addition to the product running a size small), whereas our target is a continuous fact derived from the inputs as a whole (e.g. aggregate sentiment or effect sizes). PASS is designed to produce summaries that are plausible, as opposed (and complementary) to summaries that reflect inherent contradiction in the input data. As PASS produces summaries from subsets of each instance's input, it cannot perform an explicit synthesis on its own, as opposed to our work, wherein each summary was produced with access to the whole of each instance's input.

## C  MODELS

We train all models using a modified Huggingface Transformers libraryWolf et al. (2020). For the Pegasus model, we use a distilled version provided by Huggingface (Table 7). All models were trained using their default hyperparameters, except for batch size, optimization steps, learning rates, and any parameters specified in Table 7. We fix our batch size to 16, using gradient accumulation over single instances at a time, with floating point 16 (fp16) precision (due to data size), and perform an approximate (subject to resource constraints) grid search over learning rates and training steps (Tables 8, 9), keeping the model highlighted in bold. Earlier experimentation was performed ad-hoc with Longformer and T5 models only; we found that while lower numbers of steps could perform well, they had high variance and were more sensitive to hyperparameter changes than longer runs. All training was performed on 48G NVIDIA RTX8000 GPUs, most models are unable to fit single instance gradient information into fewer than 40G, even at reduced precision.

## D  DETAILED RESULTS

**Measure Validation** As our results rely on using proxy metrics, we measure the quality of these proxies. See Figure 8 for movie meta-review sentiment correlation with human results, and Table 10 for how well the automatic significance measures correlate with the underlying truth.

**Diversity Sampling.** We include detailed results for the importance of diversity sampling; the diversity sampling procedure produces better metrics in every dimension (Table 11 top left vs. bottom left.). In the systematic reviews, most metrics drop slightly and abstention increases substantially (Table 12).

| model | steps | lr | rouge1 | rouge2 | rougeL |
|---|---|---|---|---|---|
| led | 1000 | 1e-5 | 25.09 | 8.95 | 20.14 |
| led | 1000 | 1e-6 | 25.25 | 8.33 | 19.48 |
| led | 1000 | 3e-5 | 25.33 | 8.65 | 19.87 |
| led | 1000 | 5e-5 | 25.14 | 8.35 | 19.89 |
| led | 5000 | 1e-5 | 25.46 | 8.63 | 19.76 |
| led | 5000 | 1e-6 | 25.31 | 8.76 | 20.12 |
| led | 5000 | 3e-5 | 24.50 | 7.49 | 19.02 |
| led | 5000 | 5e-5 | 23.99 | 7.07 | 17.61 |
| led | 10000 | 1e-5 | 24.28 | 7.60 | 19.25 |
| led | 10000 | 1e-6 | 25.58 | 8.64 | 20.32 |
| **led** | **10000** | **3e-5** | **25.60** | 7.97 | 19.59 |
| pegasus | 1000 | 1e-3 | 23.49 | 7.25 | 17.93 |
| pegasus | 1000 | 1e-4 | 22.25 | 7.23 | 17.67 |
| pegasus | 1000 | 1e-5 | 18.95 | 4.23 | 14.34 |
| pegasus | 1000 | 1e-6 | 18.28 | 3.10 | 13.08 |
| pegasus | 2500 | 1e-3 | 26.44 | 9.30 | 20.45 |
| pegasus | 2500 | 1e-4 | 26.50 | 10.81 | 20.91 |
| pegasus | 2500 | 1e-5 | 24.98 | 10.17 | 19.75 |
| pegasus | 2500 | 1e-6 | 23.02 | 7.92 | 18.23 |
| pegasus | 5000 | 1e-3 | 24.05 | 7.75 | 18.98 |
| pegasus | 5000 | 1e-4 | 27.41 | 10.26 | 21.72 |
| pegasus | 5000 | 1e-5 | 25.67 | 9.86 | 20.28 |
| pegasus | 5000 | 1e-6 | 23.57 | 8.74 | 18.77 |
| pegasus | 10000 | 1e-3 | 23.18 | 7.17 | 17.19 |
| **pegasus** | **10000** | **1e-4** | **27.42** | 9.53 | 21.05 |
| pegasus | 10000 | 1e-5 | 25.85 | 10.25 | 20.41 |
| pegasus | 10000 | 1e-6 | 24.41 | 9.88 | 19.72 |
| primera | 2500 | 1e-4 | 23.32 | 7.02 | 18.10 |
| primera | 2500 | 1e-5 | 25.12 | 8.39 | 19.52 |
| primera | 2500 | 1e-6 | 24.92 | 8.48 | 19.93 |
| primera | 5000 | 1e-4 | 24.35 | 7.40 | 18.49 |
| **primera** | **5000** | **1e-5** | **25.42** | 8.44 | 19.81 |
| primera | 5000 | 1e-6 | 25.32 | 8.75 | 20.06 |
| primera | 10000 | 1e-4 | 23.57 | 7.24 | 17.89 |
| primera | 10000 | 1e-5 | 24.27 | 7.59 | 18.55 |
| primera | 10000 | 1e-6 | 25.39 | 8.66 | 20.12 |
| t5 | 1000 | 1e-4 | 25.24 | 9.13 | 19.67 |
| t5 | 1000 | 1e-5 | 24.31 | 7.87 | 19.30 |
| t5 | 1000 | 1e-6 | 22.39 | 6.62 | 17.90 |
| t5 | 1000 | 5e-5 | 25.06 | 8.65 | 19.96 |
| **t5** | **2500** | **1e-4** | **25.82** | 8.46 | 19.59 |
| t5 | 2500 | 1e-5 | 24.94 | 8.36 | 19.61 |
| t5 | 2500 | 1e-6 | 23.82 | 7.59 | 19.09 |
| t5 | 2500 | 5e-5 | 25.57 | 8.47 | 19.71 |
| t5 | 5000 | 1e-4 | 25.11 | 8.17 | 19.84 |
| t5 | 5000 | 1e-5 | 25.07 | 8.57 | 19.55 |
| t5 | 5000 | 1e-6 | 23.87 | 7.99 | 19.36 |
| t5 | 5000 | 5e-5 | 24.47 | 8.40 | 19.53 |
| t5 | 7500 | 1e-4 | 24.33 | 7.58 | 18.93 |
| t5 | 7500 | 1e-5 | 25.67 | 8.75 | 19.92 |
| t5 | 7500 | 1e-6 | 24.17 | 7.73 | 19.43 |
| t5 | 7500 | 5e-5 | 25.66 | 8.64 | 19.59 |
| t5 | 10000 | 1e-4 | 24.41 | 7.78 | 18.89 |
| t5 | 10000 | 1e-5 | 25.73 | 8.91 | 20.10 |
| t5 | 10000 | 1e-6 | 24.29 | 7.98 | 19.31 |
| t5 | 10000 | 5e-5 | 25.09 | 7.89 | 19.19 |

Table 8: Movie Reviews dev training results, best models bolded.

| Model | steps | lr | ROUGE1 | ROUGE2 | ROUGEL |
|---|---|---|---|---|---|
| led | 250 | 5e-5 | 24.30 | 6.70 | 18.68 |
| led | 500 | 5e-5 | 23.99 | 6.91 | 17.09 |
| led | 1000 | 5e-5 | 25.21 | 7.31 | 18.54 |
| led | 2500 | 5e-5 | 26.05 | 6.52 | 17.65 |
| **led** | **5000** | **5e-5** | **30.96** | 8.65 | 20.33 |
| pegasus | 250 | 1e-4 | 20.22 | 5.64 | 15.86 |
| pegasus | 500 | 1e-4 | 21.66 | 6.71 | 16.92 |
| pegasus | 1000 | 1e-4 | 21.87 | 6.67 | 16.79 |
| pegasus | 2500 | 1e-4 | 22.44 | 6.71 | 17.24 |
| **pegasus** | **5000** | **1e-4** | **22.66** | 5.69 | 16.87 |
| primera | 250 | 1e-4 | 23.21 | 7.68 | 17.96 |
| primera | 500 | 1e-4 | 22.80 | 5.72 | 16.39 |
| primera | 1000 | 1e-4 | 26.08 | 6.70 | 16.77 |
| primera | 2500 | 1e-4 | 27.60 | 6.99 | 18.16 |
| primera | 5000 | 1e-4 | 27.70 | 6.96 | 18.02 |
| primera | 250 | 5e-5 | 22.52 | 6.90 | 17.98 |
| primera | 500 | 5e-5 | 24.22 | 6.11 | 17.53 |
| primera | 1000 | 5e-5 | 24.53 | 6.71 | 17.01 |
| primera | 2500 | 5e-5 | 27.64 | 6.89 | 18.79 |
| **primera** | **5000** | **5e-5** | **28.46** | 6.59 | 18.08 |
| **t5** | **250** | **5e-5** | **23.80** | 7.09 | 18.58 |
| t5 | 500 | 5e-5 | 22.77 | 6.99 | 18.30 |
| t5 | 1000 | 5e-5 | 0.00 | 0.00 | 0.00 |
| t5 | 2500 | 5e-5 | 0.00 | 0.00 | 0.00 |

Table 9: Systematic Reviews dev training results, best models bolded. We experimented with other parameters (in particular learning rates), and found that total number of steps was more important.

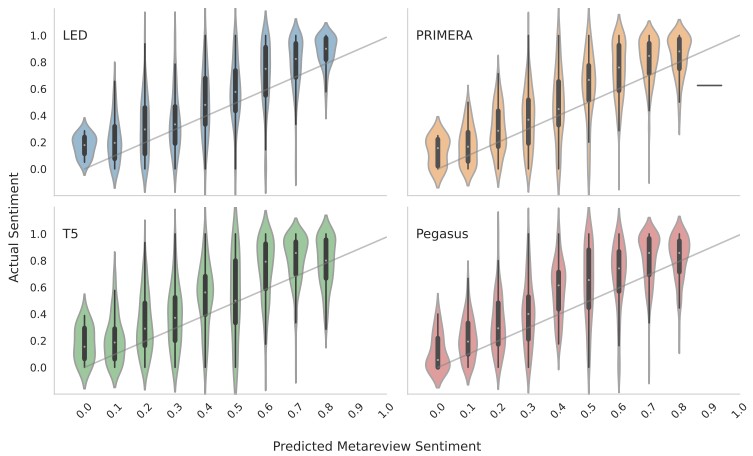

Figure 8: Actual sentiment vs. predicted sentiments on model outputs.

# E ROUGE RESULTS

We report mean differences in ROUGE outputs for both datasets in Figure 10. Ideally, these would have all mass at zero.

| | Precision | Recall | F1-score | Support |
|---|---|---|---|---|
| No significant difference | 0.726 | 0.870 | 0.792 | 247 |
| Significant difference | 0.500 | 0.283 | 0.362 | 113 |
| Accuracy | | | 0.686 | 360 |
| Macro avg | 0.613 | 0.577 | 0.577 | 360 |

Table 10: Systematic review significance validation results.

| | $R^2$ | MSE | Pearson's r | R1 | | $R^2$ | MSE | Pearson's r | R1 |
|---|---|---|---|---|---|---|---|---|---|
| LED | 0.656 | 0.032 | 0.821 | 0.229 | LED | 0.711 | 0.027 | 0.847 | 0.240 |
| Pegasus | 0.694 | 0.029 | 0.835 | 0.229 | Pegasus | 0.705 | 0.028 | 0.840 | 0.247 |
| PRIMERA | 0.749 | 0.024 | 0.880 | 0.240 | PRIMERA | 0.731 | 0.025 | 0.857 | 0.255 |
| T5 | 0.721 | 0.026 | 0.856 | 0.231 | T5 | 0.669 | 0.031 | 0.819 | 0.253 |
| Reference | 0.697 | 0.023 | 0.836 | | Reference | 0.697 | 0.023 | 0.836 | |
| | $R^2$ | MSE | Pearson's r | R1 | | $R^2$ | MSE | Pearson's r | R1 |
| LED | 0.653 | 0.033 | 0.815 | 0.241 | LED | 0.763 | 0.022 | 0.878 | 0.227 |
| PEGASUS | 0.649 | 0.033 | 0.809 | 0.248 | Pegasus | 0.799 | 0.019 | 0.894 | 0.232 |
| PRIMERA | 0.685 | 0.029 | 0.833 | 0.254 | PRIMERA | 0.890 | 0.011 | 0.948 | 0.240 |
| T5 | 0.615 | 0.036 | 0.786 | 0.252 | T5 | 0.876 | 0.012 | 0.938 | 0.230 |
| Reference | 0.697 | 0.023 | 0.836 | | Reference | 0.697 | 0.023 | 0.836 | |

Table 11: Movie Reviews. Top left: Generate 5 diverse movie meta-reviews and then choose among them using an approximate target sentiment. Top right: Generate 25 diverse movie meta-reviews and then choose among them using an approximate target sentiment; this was accidentally referenced in an earlier version of this work. Bottom left: Generate 5 movie meta-reviews using standard beam search and choose among them using an approximate target sentiment. Bottom right: Generate 5 diverse movie meta-reviews and select amongst them using the oracle sentiment. In all cases R1 refers to ROUGE1.

| | F1 | Abstention | ROUGE1 | Abstention-Oracle | ROUGE1-Oracle |
|---|---|---|---|---|---|
| LED | 0.521 | 0.503 | 0.258 | 0.358 | 0.263 |
| PRIMERA | 0.551 | 0.464 | 0.256 | 0.342 | 0.248 |
| T5 | 0.546 | 0.422 | 0.204 | 0.328 | 0.211 |
| Pegasus | 0.589 | 0.469 | 0.211 | 0.281 | 0.222 |
| Reference | 0.577 | | | | |

Table 12: Systematic reviews results with multiple generate-then-select predictions, this time using the top-5 results from standard beam-search. F1 is a macro-averaged F1 on the set of returned results. We abstain when no output matches the expected synthesis result.

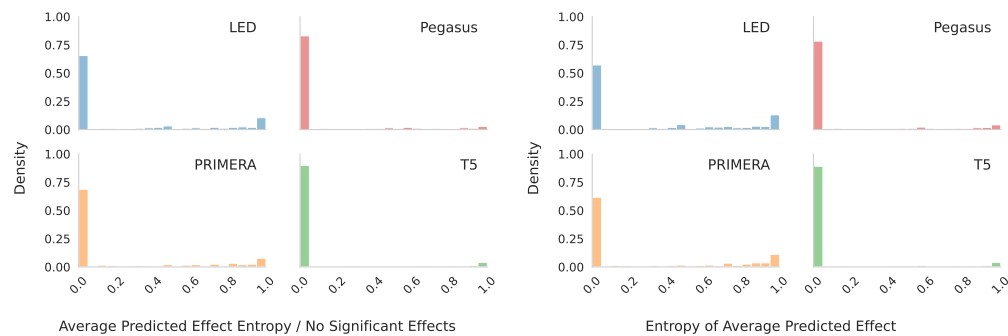

Figure 9: Entropy of instance predictions. Broken out by whether or not the underlying truth is *not significant* (left); or the whole dataset (right)

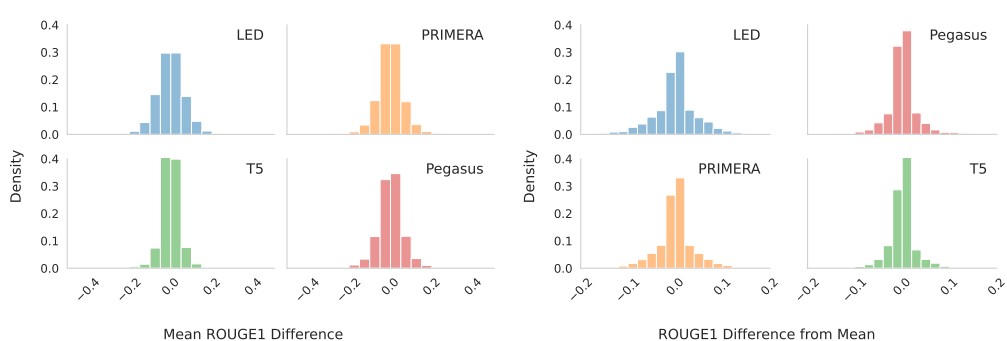

Figure 10: ROUGE1 deviations from instance means for movie reviews (left) and systematic reviews (right).

## F  EXAMPLES OF DIVERSE MOVIE SUMMARIES

| Summary | Sentiment |
|---|---|
| The Private Lives of Pippa Lee relies on a strong ensemble cast to deliver witty and poignant observations about life and relationships. | 0.800731 |
| The Private Lives of Pippa Lee relies on a strong ensemble cast to deliver witty and poignant observations about life and relationships. | 0.800731 |
| With a strong cast and Robin Wright Penn's sharp performance, The Private Lives of Pippa Lee succeeds as both a witty tribute to lost characters and a showcase for Robin Wright Penn. | 0.809596 |
| With a strong cast and Robin Wright Penn's empathetic direction, The Private Lives of Pippa Lee succeeds as both a humorous look at domestic issues and a poignant look at relationships. | 0.809081 |
| The Private Lives of Pippa Lee benefits from Robin Wright Penn's superb performance, as well as a strong ensemble cast that includes Keanu Reeves, and Faye Dunaway. | 0.845693 |
| The Private Lives of Pippa Lee has an affecting ensemble cast and Robin Wright Penn delivers a noteworthy performance, although the film is a bit too episodic. | 0.654905 |

Table 13: Different meta-reviews of "The Private Lives of Pippa Lee" and corresponding sentiments. The target sentiment for this meta-review is 70%, generating diverse candidates helps find a meta-review closer to the target.

| Summary | Sentiment |
| --- | --- |
| You Don't Mess With the Zohan's handful of laughs are almost enough to compensate for its inconsistent tone and stale, obvious jokes. | 0.242698 |
| You Don't Mess with the Zohan has a handful of crotch thrusts, but not enough of them land. | 0.429654 |
| You Don't Mess With the Zohan's handful of laughs are almost enough to compensate for its aimless, crass script. | 0.287896 |
| You Don't Mess with the Zohan has its moments, but not all of them – and the jokes are embarrassingly crass and often crude. | 0.434442 |
| You Don't Mess with the Zohan has its moments, but not all of them – and the jokes are embarrassingly crass and often crude. The script | 0.406172 |

Table 14: Different meta-reviews for "You Don't Mess With The Zohan"; a relatively panned movie with a target meta-review sentiment of 37%.

## G    EXAMPLES OF DIVERSE SYSTEMATIC REVIEWS

| Generated | Effect |
| --- | --- |
| Ketanserin versus placebo in the Raynaud's phenomenon is neither effective nor safe. The Raynaud's phenomenon is associated with significant adverse effects including dizziness and pain. The effectiveness of ketanserin for the Raynaud's phenomenon is unknown. | no significant difference |
| Ketanserin versus placebo in the Raynaud's phenomenon is neither effective nor safe. The Raynaud's phenomenon is associated with significant adverse effects including dizziness and pain. | no significant difference |
| Ketanserin and serotonin receptor antagonists in the Raynaud's phenomenon treatment of systemic scleroderma reduce the incidence of ischaemic ulcers and may reduce the frequency of adverse events. | significant difference |
| The Raynaud's phenomenon is associated with a small number of adverse effects when administered orally to patients with Raynaud's phenomenon. The frequency of Raynaud's phenomenon is similar to that of other drugs. However, there is little evidence to aid the treatment of Raynaud's phenomenon. | no significant difference |
| The Raynaud's phenomenon is associated with a small number of adverse effects when administered orally to patients with Raynaud's phenomenon. The frequency of Raynaud's phenomenon is similar to that of other drugs. | no significant difference |

Table 15: An instance where generating multiple reviews allows our models to find a candidate summary reporting a significant difference (the target).

| Generated | Effect |
|---|---|
| The overall evidence supports the use of topical antibiotics in surgical patients who have undergone minor surgery, compared to no treatment. The effect on other outcomes, other than infection rate, is consistent. The safety profile of topical antibiotics is also of concern. Further well-designed RCTs are needed to assess effectiveness of topical antibiotics in surgical patients. | no significant difference |
| A single application of topical antibiotics in surgical site wounds reduces the risk of infection, and the risk of other complications, including wound dehiscence. The risk of infection recurrence is low. The use of topical antibiotics outside of surgery should be restricted to surgical site wounds. | no significant difference |
| A single application of topical antibiotics in surgical site wounds reduces the risk of infection, and the risk of other complications, including wound dehiscence. The risk of infection recurrence is low. | no significant difference |
| The overall evidence supports the use of topical antibiotics in surgical patients to reduce the risk of infection, and the risk of other complications, especially in high-risk patients. There is a lack of evidence in low-risk patients to support the use of topical antibiotics in this setting. | significant difference |
| A single application of topical antibiotics in surgical site infection prevention has been demonstrated to reduce the risk of infection in patients who have undergone surgery. The number of patients who have been treated with topical antibiotics has been small but this is due to risk of bias in the trials. Ointment use should be limited to patients whose primary wound is irradiated. | significantly difference |

Table 16: An instance where generating multiple reviews allows our models to find a candidate summary reporting a significant difference (the target).

