# OpenReview forum: "Do Summarization Models Synthesize?"
_ICLR.cc/2023/Conference — Submitted to ICLR 2023_

### Official Review · Reviewer_YA3t · 2022-10-23

**Confidence:** 4
**Correctness:** 2
**Technical Novelty And Significance:** 2
**Empirical Novelty And Significance:** 2
**Recommendation:** 3

**Clarity, Quality, Novelty And Reproducibility:**

The paper is clear and well organized. The paper can be reproduced based on the proposed method is somewhat novel and empirically supported. The quality of analysis could be improved.

**Strength And Weaknesses:**

Strengths:
- The paper tackles an important problem and proposes a simple post-hoc (inference time) method for improving synthesis in summarization models
- The metrics provided are automatic and rely on human-written summaries, making the evaluation strong/fair

Weaknesses:
- The ordering analysis in section 5.2 could result in different results if we augmented the training data by switching the order of the reviews for example.
- Likewise, the analysis in section 5.3 addresses something that the models were not trained for. The results could see a shift if there is data augmentation with the expected result as the label.
- The abstention percentage is between 30 and 40%, which is quite high and shows the need for a change in training rather than inference.

**Summary Of The Paper:**

This paper proposes to tackle the problem of multi-document summarization through synthesis evaluation. Do summarization models synthesize large and varied inputs? The paper studies this question from two angles: movie reviews, where the meta-review is human-written and scores are given (roughly equivalent to sentiment scores); and biomedical systematic reviews of treatments, where multiple systematic reviews summarize clinical trials, resulting in statistical meta-analysis results that can be used as scores. The authors make a good choice of pre-trained models to analyze and show that they do not synthesize a large body of input as well. Then they further analyze how the models fare when we switch the input ordering, or the balance of the input (by ablation). Finally, they propose their own post-hoc method consisting of generating a diverse set of output candidates, and then select the set that best corresponds to an expected synthesis result obtained using an aggregate score predicted by an external pre-trained model.

**Summary Of The Review:**

While the authors propose an empirically-supported method to improve synthesis in summarization models, their analysis is lacking as they have evaluated models against input ordering and re-balancing, which the models could be trained for but were not. The result is a good contribution, but the analysis suggests that a training-time method can improve the scores further. The high rate of abstention is another flaw that hints at a required change in training rather than inference.

---

> ### Author Response · Authors · 2022-11-17
> **Response to Reviewer YA3t Part 3 (abstention concerns)**
>
> > The abstention percentage is between 30 and 40%, which is quite high and shows the need for a change in training rather than inference.
>
> We agree that this is high (meaning that the model fails to produce a summary which seems to agree with the anticipated synthesized aspect of interest). Indeed, in our view this highlights the need for new training strategies which address this issue of synthesis in summarization, and our hope is that this initial work motivates such approaches.
>
> Nonetheless, with current techniques (those explored), we would argue it may still be preferable to abstain in such cases; even if this means abstaining often. For example, providing a misleading summary of medical evidence would be worse than providing no summary at all.

---

> ### Author Response · Authors · 2022-11-17
> **Response to Reviewer YA3t Part 2 (augmentation strategy)**
>
> > Likewise, the analysis in section 5.3 addresses something that the models were not trained for. The results could see a shift if there is data augmentation with the expected result as the label.
>
> Models here are trained to produce summaries of multiple input (movie reviews or clinical trial abstracts). Changing input compositions (e.g., removing some fraction of positive reviews for a particular instance) does not change this task, and the model will have encountered instances which feature a full range of positive:negative ratios in training. Therefore, models should be capable of producing summaries of critiques (or trial reports) whatever their composition with respect to sentiment (or effect sizes). We therefore disagree that we are asking the model to do something the models were “not trained for”.
>
> In any case, augmenting the data in training by manipulating input compositions is not possible because this would require associated training targets (summaries) that reflect the corresponding altered composition. For example, if during training we provide the model with a (subset of) reviews that are mostly negative for a film which in fact received mixed reviews, we would need an associated negative target summary, which would have to be manually composed. This augmentation strategy would therefore require (substantial) additional supervision.
>
>
> Continuing the results from above, we report results using models trained with reshuffled input orderings on changing aggregate input characteristics (i.e. sentiment and effect size):
>
> | Training     | Model      | F1-Score | Accuracy |
> | -- | -- | -- | -- |
> | Original     | LED    | 0.510          | 0.684 |
> | Original     | PRIMERA| 0.533      | 0.680 |
> | Shuffled    | LED        | 0.522       | 0.680 |
> | Shuffled    | PRIMERA | 0.552    | 0.684 |
>
> Systematic Review Sensitivity: model sensitivity to input changes show slight improvements when changing from a fixed (but random) training order to a constantly variable training order.
>
> | Training        | name        | $R^2$ | Pearson’s r | MSE        |
> | -- | -- |  -- | -- | -- |
> | Orignal    | PRIMERA      | .572    |  .756            |   .052               |
> | Original    |         T5      | .481    |  .694            | .063                 |
> | Shuffled    | PRIMERA      | .575    | .758                | .051              |
> | Shuffled    |         T5      | .495    |  .703             |.061            |
>
> Movie meta review synthesis results: model sensitivity to input changes again shows slight improvements when changing from a fixed (but random) training order to a constantly variable training.
>
> We will complete this experiment (and those corresponding to section 5.2) across all models and report the results in an appendix entry regarding amended training. There is not enough time before the deadline to complete this experiment for all four model architectures.

---

> ### Author Response · Authors · 2022-11-17
> **Response to Reviewer YA3t Part 1 (training strategies and augmentation)**
>
> We thank reviewer YA3t for their detailed feedback, and are glad they agree that this is an important area and task.
>
> > The ordering analysis in section 5.2 could result in different results if we augmented the training data by switching the order of the reviews for example.
>
> Prior to training models, we shuffled the input orderings for all instances, i.e., for each movie meta-review, we shuffled the order of the input reviews, and for each systematic review, we shuffled the order of the input studies. This means the order carries no signal by construction. However, as it is fixed during training; perhaps the “augmentation” strategy suggested may therefore yield benefits.
>
> To explore this, we completed an additional experiment wherein we take two models for each dataset (LED and PRIMERA for systematic reviews; PRIMERA and T5 for movie reviews) and perform augmented training as suggested, including instances redundantly with input orderings shuffled at random. Across input order perturbations, we compare the distributions of ROUGE results, whether or not the synthesis statistics (macro-f1 for systematic reviews, correlation for movie reviews) are different, and whether or not the subsampling results differ (see reply regarding section 5.3).
>
> This approach did not qualitatively change our results. Instead we observe small increases and decreases across all metrics under the proposed augmentation strategy. We report these results below, and will add these to the paper in an updated draft (along with analogous results for all model variants, which we did not have time to complete during the rebuttal window).
>
> ROUGE: we report ROUGE1 mean and standard deviations across shuffles for our models. Note results in Table 2 were reported over a single fixed ordering of the dev set, whereas these results are reported over all 100 shuffles.
>
> | Training     | Model      | Mean | Std |
> | -- | -- | -- | -- |
> | Original     |  LED        |  .262 | .0025 |
> | Original    | PRIMERA | .253 | .0021 |
> | Shuffled    |  LED        |  .257 | .0020 |
> | Shuffled    | PRIMERA | .249 | .0018 |
>
> Systematic Review ROUGE1 distributions: we see a slight decrease in average ROUGE1 result between versions trained with a static (but arbitrary) input ordering, and versions with every step shuffled.
>
> | Training     | Model         | Mean | Std |
> | -- | -- | -- | -- |
> | Original     |  PRIMERA | .251    | .0020 |
> | Original     |  T5        | .254    | .0015 |
> | Shuffled    |  PRIMERA |  .255   | .0019 |
> | Shuffled    |  T5        | .252    | .0015 |
>
> Movie Metareview ROUGE1 distributions: we see a mixed result in average ROUGE1 result between versions trained with a static (but arbitrary) input ordering, and version with every step shuffled.
>
>
> Synthesis: We report measures synthesis performance: correlation for movie meta-reviews, and macro-averaged F1s for systematic reviews
>
> | Training     | Model      | Min   | Mean | Max | Std |
> | -- | -- | -- | -- | -- | -- |
> | Original     |  LED        | .450 | .505 | .561 | .020 |
> | Original    | PRIMERA | .503 | .550 | .597 | .019 |
> | Shuffled    |  LED        | .485 | .523 | .579 | .018 |
> | Shuffled    | PRIMERA | .517 | .552 | .601 | .017 |
>
> Systematic Review Macro F1 distributions: we again see a slight increase when reshuffling instance inputs every training step.
>
> | Training        | name        | $R^2$ | Pearson’s r | MSE        |
> | -- | -- | -- | -- | -- |
> | Orignal    | PRIMERA      | .608    | .780             | .037 |
> | Original    |         T5      | .516    | .720             | .046 |
> | Shuffled    | PRIMERA      | .603    | .777             | .037 |
> | Shuffled    |         T5      | .504    | .713             | .046 |
>
> Movie meta review synthesis results: we see a slight decrease in synthesis results when reshuffling instance inputs every training step.

---

### Official Review · Reviewer_18KJ · 2022-10-24

**Confidence:** 3
**Correctness:** 3
**Technical Novelty And Significance:** 2
**Empirical Novelty And Significance:** 3
**Recommendation:** 5

**Clarity, Quality, Novelty And Reproducibility:**

The paper is clear and easy to follow. I especially liked the technical accuracy, being neither too complex nor sparse. The level of detail for the implementation is solid and generally seems reproducible. The authors don't provide the source code, which would greatly improve reproducibility (and speed up subsequent research).
The novelty is limited, the solution to address the weakness of the models is an application of existing methods.

Baseline/Related Work
- Shah et al. (2021a) solves a similar task, in a different application area. The authors need to state the technical relation to this paper in more detail. It seems to be a valid, stronger baseline than SoA summarization models not focusing on factors, and I am wondering why it hasn’t been used in the experiments (source code is available).

Experiments / Open Questions
- How the cases when the model abstained from a decision (Table 5) counted in the evaluation (e.g., Figure 7)?
- If the related work cited in footnote 2 is complementary, a remark on why that’s the case and why it doesn’t serve as a baseline would be interesting. Otherwise, this raises the question of whether the right baseline has been chosen.

**Strength And Weaknesses:**

Strengths:
[S1] The paper extends the conceptualization of desiderate for multi-document summaries to synthetization.
[S2] The paper is well written and methodological sound.
[S3] The proposed measures, given that factor information is available, could be used as evaluation metric for MDS tasks.

Weaknesses:
[W1] The novelty of the method is limited.
[W2] Some questions in the design and results of the experiment remain open.
[W3] A stronger baseline could have been used.

**Summary Of The Paper:**

The paper investigates the synthesizing capabilities of multi-document summarization models and presents a method for increasing those capabilities in summarization.
Synthesizing capabilities are measured with respect to an application-dependent latent factor, the model is required to generate outputs that align with that factor in the input. Two cases have been studied: movie reviews with the latent factor of sentiment, and medical study results with the latent factor of significant effects for interventions. The investigated models proved to be lacking in synthesizing capabilities, but the proposed method raised these to human level (with regard to the metrics).

**Summary Of The Review:**

The paper addresses an important issues for multi-document summarization, and show empirical evidence that this issue persists in SoA summarization models. Conceptualisation and experiments are generally sound. It misses a detailed discussion of one highly relevant work, and it's inclusion in the comparison. The paper is generally reproducible from the given details, inclusion of source code would make more so.

---

> ### Author Response · Authors · 2022-11-14
> **Response to Reviewer 18KJ**
>
> We thank Reviewer 18KJ for their detailed feedback. We emphasize our main contribution is empirically demonstrating that summarization models do not  reliably perform synthesis implicitly; we are glad that the reviewer also thinks this an important problem. Our hope is that this empirical analysis and framing motivates work on improving multi-document summarization models and their ability to synthesize; we view the methodological contribution here as secondary. We also appreciate the feedback about the technical balance; we found this hard to achieve and are pleased we managed.
>
> > How the cases when the model abstained from a decision (Table 5) counted in the evaluation (e.g., Figure 7)?
>
> In Table 5 the results are reported over the set of reviews where the predicted synthesis aspect matched any of the candidate summaries. Figure 7 (right) is reported over the same summaries selected in Table 5.
>
>
> > If the related work cited in footnote 2 is complementary, a remark on why that’s the case and why it doesn’t serve as a baseline would be interesting. Otherwise, this raises the question of whether the right baseline has been chosen.
>
> We agree with the reviewer that we should have provided additional details concerning Shah et al.’s work on the “nutri-bullets” system (which does also address the broader problem of generating “consensual” summaries) and how our work differs from this in assumptions and approach. In brief, Shah et al. assume a low-supervision setting (with respect to input/output pairs) in which one has a set of tuples extracted from texts with which to train a content extractor, and where one can design heuristic rule-based aggregation strategies on top of extracted tuples mapping onto discrete categories like “consensus”. By contrast, we assume a more typical supervised summarization setting and are interested in continuous aggregation of a latent attribute of interest, and we do not assume (or have access to) relational tuples over inputs.
>
> First, Shah et al. are focussed primarily on settings in which training data is (severely) limited, and motivate their pipeline approach on the basis of this limited supervision assumption. For this reason they define separate modules: The first performs content selection (tuple extraction; this does require manual annotations of tuples on a subset of texts to train such an extractor); The second applies (manually composed) deterministic aggregation rules on top of these extracted tuples to combine them; A final module then generates a “surface realization” conditioned on the aggregated result.
>
> We have investigated more typical supervised settings (with thousands of input and summary pairs), and we are training modern end-to-end transformer-based summarization models. We have empirically assessed the extent to which model outputs in this typical training regime are consistent with the continuous synthesis result anticipated. We do not have annotated tuples on our inputs (as would be required to use the Shah et al. approach, as it assumes a trained content extractor module). And while applying discrete (manually composed) aggregation operators over inputs makes sense in some settings, we are explicitly interested in the ability of models to aggregate variables of interest continuously, for example producing “very positive” summaries when movie reviews are overwhelmingly positive, and merely “positive” summaries when they are only mostly positive.
>
> In sum, the approach proposed by Shah et al. is appropriate in, and designed for, low-supervision settings (which we do not consider here) where there are natural “tuples” to be extracted from inputs and supervision for this sub-task (which we do not have) and where discrete aggregation categories of inputs is natural (whereas we are interested in continuous aggregation, e.g., mean sentiment). We have included this discussion in the Appendix.

---

### Official Review · Reviewer_Z9mN · 2022-10-25

**Confidence:** 2
**Correctness:** 3
**Technical Novelty And Significance:** 2
**Empirical Novelty And Significance:** 3
**Recommendation:** 5

**Clarity, Quality, Novelty And Reproducibility:**

This paper contains originality because this paper is the first study to investigate synthesis in multi-document summarization.

**Strength And Weaknesses:**

Strengths:

- This paper tackles one of the important issues in multi-document summarization: the existing multi-document summarizers can synthesize inputs properly.
- This paper includes extensive analysis with statistical results on two data sets.

Weaknesses:

- The main criticism is that the diverse beam search (DBS) does not generate diverse candidates with various ``properties''. The DBS generates a set of candidates by penalizing the generation of strings similar to candidates on beams in other groups. In DBS, the similarity is only considered at the token level, not the property of interest level. DBS is not guaranteed to generate diverse candidates with various properties of interest.
- The proposed strategy 'generate-diverse-then-select' should be compared with the strategy 'generate-then-select.' This paper has no empirical studies to show the superiority of the 'generate-diverse-then-select.'
- There is no report of the ROUGE1 score on movie review experiments (Table 4). The main goal of the proposed strategy is to select a summarization that aligns with the expected property of inputs and the target summary. Only showing $R^2$, MSE, and Pearson's r results cannot prove the superiority of the proposed method because it selects the wrong summarization but aligns with the expected property.
- The proposed method can do a cautious summarization, but there is no evaluation of the cautionary of the summaries. Human evaluations or correlations of automatic factuality metrics should be included.
- There are two versions to predict sentiment on movie data set: BERT trained on the SST data set, and BERT trained on the IMDB data set. It is a wonder why this paper adopted different models.
- Some evaluation settings are missing, like the number of beams. Furthermore, selecting the true sentiment in Table 4 is unclear. Please add the details to the appendix.
- As mentioned in the introduction, the problem considered in this paper is only for some applications, such as movies or medical, not all applications for multi-document summarization, like news. The reviewer thinks the title is too generalized for what the paper considers.

**Summary Of The Paper:**

This paper investigates the synthesis problem in multi-document summarization. Then, this paper proposes a simple method: Given a set of documents, the proposed method generates diverse candidates using the diverse beam search and selects one candidate that aligns with the expected aggregate property of inputs.

**Summary Of The Review:**

The paper shows insights about the synthesis of the existing multi-document summarizers. However, the experiments lack to prove the superiority of the proposed method.

---

> ### Author Response · Authors · 2022-11-10
> **Response to Reviewer Z9mN, Part 1, Beam Search**
>
> We thank Reviewer Z9mN for their detailed feedback. We emphasize our main contribution is empirically demonstrating that summarization models (trained in the standard way) do not  reliably perform synthesis implicitly; we are glad that the reviewer also thinks this an important problem. Our hope is that this empirical analysis and framing motivates work on improving multi-document summarization models and their ability to synthesize; we view the methodological contribution here as secondary. The reviewer raises valid issues which we feel owe mostly to presentation problems (and some missing experimental results which we have now added), and we address below.
>
>
> > The main criticism is that the diverse beam search (DBS) does not generate diverse candidates with various ``properties''. The DBS generates a set of candidates by penalizing the generation of strings similar to candidates on beams in other groups. In DBS, the similarity is only considered at the token level, not the property of interest level. DBS is not guaranteed to generate diverse candidates with various properties of interest.
>
> DBS generates summaries that are diverse with respect to the tokens they can contain, but which are nonetheless all plausible. This is a mechanism for generating candidates which are different from one another, and the idea is that this diversity with respect to token choice is likely to correspond to differences with respect to properties of interest. Our empirical results suggest this is indeed the case (e.g., see Figure 7).
>
> > The proposed strategy 'generate-diverse-then-select' should be compared with the strategy 'generate-then-select.' This paper has no empirical studies to show the superiority of the 'generate-diverse-then-select.'
>
> We completely agree that including a “generate-then-select” approach is an important ablation  missing from the original submission. We have updated our draft to include these experiments. In short, explicitly generating diverse candidates indeed results in improved synthesis performance. One can see this in Tables 11 and 12 on page 17 in the current draft (partially reproduced below).
>
> |         | R$^2$ | MSE   | Pearson's r | R1    |
> |-----------|-------|-------|-------------|-------|
> | LED       | 0.656 | 0.032 | 0.821       | 0.229 |
> | Pegasus   | 0.694 | 0.029 | 0.835       | 0.229 |
> | PRIMERA   | 0.749 | 0.024 | 0.880       | 0.240 |
> | T5        | 0.721 | 0.026 | 0.856       | 0.231 |
> | Reference | 0.697 | 0.023 | 0.836       |       |
>
> Table 11, upper left, generate 5 diverse movie meta-reviews and then choose among
> them using an approximate target sentiment.
>
> |         | R$^2$ | MSE   | Pearson's r | R1    |
> |-----------|-------|-------|-------------|-------|
> | LED       | 0.653 | 0.033 | 0.815       | 0.241 |
> | PEGASUS   | 0.649 | 0.033 | 0.809       | 0.248 |
> | PRIMERA   | 0.685 | 0.029 | 0.833       | 0.254 |
> | T5        | 0.615 | 0.036 | 0.786       | 0.252 |
> | Reference | 0.697 | 0.023 | 0.836       |
>
> Table 11, lower left, generate 5 movie meta-reviews using standard beam search and choose among them using an approximate target sentiment.
>
> Note that Table 11 (upper right, not produced here) contains a mistake we found between submission and review release: our evaluation in for movie meta-reviewsthat instance had been conducted over 25 diverse meta-reviews (25 total beams between 5 beam groups); this is the only instead we ever experimented with a beam width of more than 5 before deciding that a smaller beam was essential for comparability.
>
>
> |         | F1    | Abstention | ROUGE1 | Abstention-Oracle | ROUGE1-Oracle |
> |-----------|-------|------------|--------|-------------------|---------------|
> | LED       | 0.521 | 0.503      | 0.258  | 0.358             | 0.263         |
> | PRIMERA   | 0.551 | 0.464      | 0.256  | 0.342             | 0.248         |
> | T5        | 0.546 | 0.422      | 0.204  | 0.328             | 0.211         |
> | Pegasus   | 0.589 | 0.469      | 0.211  | 0.281             | 0.222         |
> | Reference | 0.577 |            |
>
> Table 12. Systematic reviews results with multiple generate-then-select predictions, this time using the top-5 results from standard beam-search. F1 is a macro-averaged F1 on the set of returned results. We abstain when no output matches the expected synthesis result.

---

> ### Author Response · Authors · 2022-11-10
> **Response to Reviewer Z9mN, Part 2, ROUGE Results**
>
> > There is no report of the ROUGE1 score on movie review experiments (Table 4). The main goal of the proposed strategy is to select a summarization that aligns with the expected property of inputs and the target summary. Only showing, MSE, and Pearson's r results cannot prove the superiority of the proposed method because it selects the wrong summarization but aligns with the expected property.
>
> We agree that also considering ROUGE scores may be useful, and have added ROUGE1 scores in Table 4. When contrasting standard summarization methods (Table 2) with the generate-then-select variants (Table 4), our results suggest that alternative selection methods cause a small decrease in ROUGE1 score, while greatly increasing synthesis performance.
>
>
>
>
> |           | R$^2$       | Pearson's r | MSE         | ROUGE1 |
> |-----------|-------------|-------------|-------------|--------|
> | LED       | 0.551       | 0.742       | 0.042       | 0.242  |
> | PRIMERA   | 0.608       | 0.780       | 0.037       | 0.254  |
> | T5        | 0.516       | 0.720       | 0.046       | 0.253  |
> | Pegasus   | 0.530       | 0.730       | 0.044       | 0.245  |
> | Reference | 0.697  | 0.836 |  0.023 |        |
>
> Table 2. Base synthesis results. Movie reviews: correlations between sentiment measured in model outputs and target sentiments. We report R$^2$, Pearson's r, and mean-squared errors.
>
>
> |         | R$^2$ | MSE   | Pearson's r | ROUGE1    |
> |-----------|-------|-------|-------------|-------|
> | LED       | 0.656 | 0.032 | 0.821       | 0.229 |
> | Pegasus   | 0.694 | 0.029 | 0.835       | 0.229 |
> | PRIMERA   | 0.749 | 0.024 | 0.880       | 0.240 |
> | T5        | 0.721 | 0.026 | 0.856       | 0.231 |
> | Reference | 0.697 | 0.023 | 0.836       |       |
> |         | R$^2$ | MSE   | Pearson's r | ROUGE1    |
> | LED       | 0.763 | 0.022 | 0.878       | 0.227 |
> | Pegasus   | 0.799 | 0.019 | 0.894       | 0.232 |
> | PRIMERA   | 0.890 | 0.011 | 0.948       | 0.240 |
> | T5        | 0.876 | 0.012 | 0.938       | 0.230 |
> | Reference | 0.697 | 0.023 | 0.836       |
>
> Table 4. Movie Reviews: Generate diverse movie meta-reviews and then choose among them using an approximate target sentiment (top) or the oracle sentiment (bottom).
>
> For reference, Table 8 in the appendix contains ROUGE1 results from fine-tuning and hyperparameter searches, and Figure 10 includes a variance in ROUGE1 scores.

---

> ### Author Response · Authors · 2022-11-10
> **Response to Reviewer Z9mN, Part 3, Cautious Summarization**
>
> > The proposed method can do a cautious summarization, but there is no evaluation of the cautionary of the summaries. Human evaluations or correlations of automatic factuality metrics should be included.
>
> We do not have a human evaluation of the cautionary summaries, however we do have automated metrics comparing the cautionary scores against non-cautionary instances. Contrasting Table 2 (right) (raw measure for synthesis quality) and Table 5 (improvements via generate-diverse-then-select) shows an improvement in overall F1 at the expense of abstaining in over a third of the instances.
>
> |         | F1-score    | ROUGE1 |
> |-----------|-------------|--------|
> | LED       | 0.490       | 0.259  |
> | PRIMERA   | 0.526       | 0.253  |
> | T5        | 0.521       | 0.206  |
> | Pegasus   | 0.568       | 0.212  |
> | Reference | {\bf 0.577} |        |
>
> Table 2 (right). Macro-averaged F1s and ROUGE1s for systematic reviews.
>
>
> |        | F1    | \%Abstention | ROUGE1 | Abstention-Oracle | ROUGE1-Oracle |
> |-----------|-------|--------------|--------|-------------------|---------------|
> | LED       | 0.557 | 0.386        | 0.252  | 0.233             | 0.259         |
> | PRIMERA   | 0.581 | 0.336        | 0.251  | 0.213             | 0.248         |
> | T5        | 0.568 | 0.350        | 0.202  | 0.228             | 0.210         |
> | Pegasus   | 0.588 | 0.383        | 0.211  | 0.242             | 0.225         |
> | Reference | 0.577 |              |
>
> Table 5. Systematic Review results with modified-then-selected predictions. F1 is a macro-averaged F1 on the set of returned results. We abstain when no output matches the expected synthesis result.
>
>
> As for the quality of our automated metrics, we report those for systematic reviews in Table 10. Overall we would describe the quality of the systematic review metrics as noisy, but sufficient to recognize strong effects, or in this case, the relative inability of the underlying models to handle synthesis.
>
>  |                         | Precision | Recall | F1-score | Support |
> |---------------------------|-----------|--------|----------|---------|
> | No significant difference | 0.726     | 0.870  | 0.792    | 247     |
> | Significant difference    | 0.500     | 0.283  | 0.362    | 113     |
> | Accuracy                  |           |        | 0.686    | 360     |
> | Macro avg                 | 0.613     | 0.577  | 0.577    | 360     |
>
> Table 10. Systematic review significance validation results.

---

> ### Author Response · Authors · 2022-11-10
> **Response to Reviewer Z9mN, Part 4, Experimental Details**
>
> > There are two versions to predict sentiment on movie data set: BERT trained on the SST data set, and BERT trained on the IMDB data set. It is a wonder why this paper adopted different models.
>
> The difference here are the targets; one is binary (classification) and one is more granular (ordinal). Specifically, the BERT-model trained on the SST-dataset is a continuous model for evaluating the sentiment in meta-reviews. The BERT-model trained on the IMDB-dataset is used to mimic the process of creating the Rotten Tomatoes meta-reviews: each of the input movie reviews is given a binary polarity (our prediction target), and the reviews are aggregated to create a meta-review for the movie. So the SST-BERT provides a continuous measure of sentiment,  while IMDB-BERT is a binary classification model. The latter is used solely for input-selection in the generate-then-select cases, switching from the IMDB-BERT to the SST-BERT or a classifier trained on this data will not provide additional information over the Oracle ablation.
>
>
> > Some evaluation settings are missing, like the number of beams. Furthermore, selecting the true sentiment in Table 4 is unclear. Please add the details to the appendix.
>
> Some of these details (beams, beam groups, always equal to 5) are specified in footnote 12 on page 7. We have clarified the wording on this footnote. Additional hyperparameter settings are reported in Tables 8 and 9 for our fine-tuning experiments. If not specified, we used the Huggingface default parameters. We will release code after de-identification.
>
> We agree that using the term “true sentiment” is unclear, and have switched to referring to the underlying sentiment of movie reviews as the “oracle sentiment”. This is the fraction of input reviews that are positive.
>
> Footnote 12: This penalty is associated with a hyperparameter $\lambda$ that encodes the relative importance of realizing diverse; we use $\lambda$=0.5 here and did not extensively tune this. Other hyperparameters include number of groups and total number of beams; we used 5 for both of these, retaining 5 beams as used for analysis above.

---

### Official Review · Reviewer_ACAC · 2022-10-25

**Confidence:** 4
**Correctness:** 3
**Technical Novelty And Significance:** 3
**Empirical Novelty And Significance:** 3
**Recommendation:** 6

**Clarity, Quality, Novelty And Reproducibility:**

The paper is very clear and easy to follow. Experimental details and hyperparameters are well-defined for reproducibility. The proposed evaluation approach presents novelty but the idea of generating several candidate summaries is already explored in previous research.

**Strength And Weaknesses:**

Strengths:
1. The paper presents an interesting way to evaluate summarization that complements traditional ROUGE scores.
2. The experiments are conducted in two quite different domains.

Weaknesses:
1. It is not clear why the authors decide not to fine-tune the measurement model on the datasets, especially for Rotten Tomatoes. The BERT and RobotViewer models were trained on human-written inputs and are now used on machine generated summaries, which might detrimental for its performance (and correlations to gold sentiments).
2. The summarizers are the "base" models and the gap in quality with respect to the large versions could change some of the conclusions. At least one experiment with a large version (PEGASUS, for instance) could make the empirical evidence much stronger.
3. Citation missing: Oved and Levy (2021) also used a strategy to generate several summaries and rank them according to a desired criteria. https://aclanthology.org/2021.acl-long.30/

Minor comments:
1. Figure 10 in the Appendix is cut.

**Summary Of The Paper:**

This paper investigates the capacity of summarization models to synthesize (potentially conflicting) information from multiple documents. It defines a model-based metric for the aggregate latent aspect of interest, in this case, the sentiment of movie reviews (Rotten Tomatoes dataset) and the treatment efficacy of medical trials. This metric is applied to reference summaries as well as summaries generated by Longformer, PEGASUS, PRIMERA, and T5. Then, the results are compared to gold labels from the datasets so that a higher correlation would be an indicator for better synthesis capacity. The experimental results indicate inconsistent synthesis for off-the-shelf summarization models. The authors then propose a technique for improving model synthesis by ranking candidate summaries according to the expected aggregated aspect.

**Summary Of The Review:**

This work presents a novel way to evaluate multi-document summarization and its experimental methods are sound. I would recommend the authors to address the experimental improvements listed above to make the claims stronger.

---

> ### Author Response · Authors · 2022-11-18
> **Response to Reviewer ACAC**
>
> We thank the reviewer for their thoughtful comments, and respond to the specific critiques below.
>
> > It is not clear why the authors decide not to fine-tune the measurement model on the datasets, especially for Rotten Tomatoes. The BERT and RobotViewer models were trained on human-written inputs and are now used on machine generated summaries, which might detrimental for its performance (and correlations to gold sentiments).
>
> We intentionally choose out-of-domain (external) models to ensure our measurements were independent from the datasets used for evaluation. Fine-tuning a model on human-authored target summaries from the evaluation datasets may yield models that are particularly well calibrated on human-authored summaries in the respective domains by virtue of having been trained on such examples. Our assumption is that external models—trained on other corpora, so the specific style of reviews will differ from both the human and model generated texts we use for evaluation—are less prone to this particular kind of bias (although they may come with a larger variance in the measure).
>
> > The summarizers are the "base" models and the gap in quality with respect to the large versions could change some of the conclusions. At least one experiment with a large version (PEGASUS, for instance) could make the empirical evidence much stronger.
>
> Due to resource (memory) constraints, we have unfortunately been unable to run models substantially larger than the base configurations. Due to gradient sizes it is typically impossible to fit a single instance in memory without using floating point 16 (fp16) precision. Even using 48G NVIDIA RTX8000 GPUs, most of the base models can fit at most a single 4k token length instance in a forward pass.  We have attempted fine-tuning additional Pegasus-large variants (google/bigbird-pegasus-large-bigpatent for movie reviews, and google/bigbird-pegasus-large-pubmed for systematic reviews), completing an approximate learning rate sweep across .32 (this unusually high learning rate was used in the originating papers for fine-tuning), 1e-3, 1e-4, 1e-5, 1e-6, across differing numbers of training steps (250, 500, 1000, 2500, 5000, 10000, 20000), repeating our training procedures as for other models. The best held-out ROUGE1 result for sentiment was 23.68 for movie reviews at 20000 steps with a learning rate of 1e-4, and for systematic reviews was 19.84 at 250 steps with a learning rate of 1e-3. Given these relatively poor results (particularly in comparison to other models), we will attempt to continue fine-tuning. We believe the performance in these cases is mostly due to instability caused by using fp16 precision; an unavoidable compromise without very large-scale compute resources, which we do not have access to.
>
> Larger models, including models such as GPT-3 (including instruct variants in similar architectures), UL2, and FLAN-T5 cannot handle sequences this long due to a combination of model size and input lengths. To use a larger model, these tasks require either larger hardware, different computation approaches (we’re optimistic given recent developments for INT8 training), or different architectures.
>
> > Citation missing: Oved and Levy (2021) also used a strategy to generate several summaries and rank them according to a desired criteria. https://aclanthology.org/2021.acl-long.30/
>
> We thank the reviewer for referring us to Oved and Levy, 2021; we agree that this is relevant and we should  include a discussion of the Perturb and Selection Summarizer (PASS) system in relation to this work. PASS is for summarizing product reviews. It works by perturbing model inputs (i.e. keep random subsets of the input), generating a summary for each perturbation (via standard beam search), and then selecting amongst outputs (via a ranker) to produce a coherent, self-consistent, and fluent summary.
>
> PASS is similar to our work in that it generates multiple outputs and selects amongst them. However it differs in several key respects. The key conceptual difference between PASS and our work is that PASS’s target is a summary’s self-consistency (a product review might contradict itself on some aspect, e.g. simultaneously discussing a product fitting well in addition to the product running a size small), whereas our target is a continuous fact derived from the inputs as a whole (e.g. aggregate sentiment or effect sizes). PASS is designed to produce summaries that are plausible, as opposed (and complementary) to summaries that reflect inherent contradiction in the input data. As PASS produces summaries from subsets of each instance’s input, it cannot perform an explicit synthesis on its own, as opposed to our work, wherein each summary was produced with access to the whole of each instance’s input.
>
>
> > Figure 10 in the Appendix is cut.
>
> We have fixed this, thank you!

---

### Decision · Program_Chairs · 2023-01-20

**Decision:**

Reject

**Justification For Why Not Higher Score:**

N/A

**Justification For Why Not Lower Score:**

N/A

**Metareview: Summary, Strengths And Weaknesses:**

This paper investigates whether multi-document summarization models can effectively synthesize conflicting or diverse information from multiple documents, and further introduces a new technique for improving model synthesis.  The strengths lie in the unique way to evaluate summarization models, the focus of an important aspect of multi-document summarization, with solid experiments conducted in very different domains. The main criticisms include the baseline choices, evaluation details, and the approach not being able to generate diverse candidates, as well as the vague generalization of this method on other general multi-document summarization. Taking into account these major concerns and after reviewing the authors’ responses, I’d encourage the authors to further improve their work for a stronger future submission.

**Summary Of Ac-Reviewer Meeting:**

N/A